# Synergizing Machine Learning Algorithm with Triboelectric Nanogenerators for Advanced Self-Powered Sensing Systems

**DOI:** 10.3390/nano14020165

**Published:** 2024-01-12

**Authors:** Roujuan Li, Di Wei, Zhonglin Wang

**Affiliations:** 1Beijing Institute of Nanoenergy and Nanosystems, Chinese Academy of Sciences, Beijing 101400, China; liroujuan@binn.cas.cn; 2School of Nanoscience and Engineering, University of Chinese Academy of Sciences, Beijing 100049, China; 3School of Materials Science and Engineering, Georgia Institute of Technology, Atlanta, GA 30332-0245, USA

**Keywords:** triboelectric nanogenerator, self-powered sensor, machine learning, deep learning, algorithm

## Abstract

The advancement of the Internet of Things (IoT) has increased the demand for large-scale intelligent sensing systems. The periodic replacement of power sources for ubiquitous sensing systems leads to significant resource waste and environmental pollution. Human staffing costs associated with replacement also increase the economic burden. The triboelectric nanogenerators (TENGs) provide both an energy harvesting scheme and the possibility of self-powered sensing. Based on contact electrification from different materials, TENGs provide a rich material selection to collect complex and diverse data. As the data collected by TENGs become increasingly numerous and complex, different approaches to machine learning (ML) and deep learning (DL) algorithms have been proposed to efficiently process output signals. In this paper, the latest advances in ML algorithms assisting solid–solid TENG and liquid–solid TENG sensors are reviewed based on the sample size and complexity of the data. The pros and cons of various algorithms are analyzed and application scenarios of various TENG sensing systems are presented. The prospects of synergizing hardware (TENG sensors) with software (ML algorithms) in a complex environment and their main challenges for future developments are discussed.

## 1. Introduction

With the rapid development of the Internet of Things (IoT), large-scale sensing systems are receiving increasing attention [1]. More than ten billion sensors will be connected to IoT in the future. However, the ubiquitous sensing systems require a sustainable power supply [2]. For example, 10 billion near-field communication sensors consume up to 40 million watts of power, which means that 5 tons of coal will be burned per hour. It could cause nearly 288,000 tons of carbon dioxide emissions. The periodic replacement of power sources for ubiquitous sensing systems leads to significant resource waste. The treatment and recycling of waste batteries increase the difficulty of environmental management [3]. Human staffing costs associated with replacement also increase the economic burden. The costs associated with system construction and maintenance limit the large-scale applications of sensors [4]. Self-powered sensors based on triboelectricity overcome these limitations due to their simple structure, low cost, self-supply capability, and high performance. The synergy of TENG and artificial intelligence (AI) optimizes the performance of intelligent self-powered sensing systems.

In 2012, Wang’s team developed the triboelectric nanogenerator (TENG), which converts mechanical energy into electrical energy by electrostatic and triboelectrification effects [5]. Traditional TENG has been widely applied for energy harvesting [6,7], collecting energy from various sources, including wind, waves, human motion, etc. TENG also offers the possibility for self-powered sensors, especially suitable for applications requiring long-term, self-sustained energy supply [8]. The self-powered sensors based on triboelectricity convert the collected energy into electrical signals. The dynamic disturbance process is detected by analyzing the high-sensitivity electrical signal. In the process of data acquisition, TENGs can realize self-supply capability. This could not only effectively reduce power consumption but also be seamlessly embedded into wearable devices [9,10,11]. Depending on the state of the triboelectric interface, TENG could be divided into solid–solid TENG (S-S TENG), liquid–solid TENG (L-S TENG), gas–solid TENG (G-S TENG), gas–liquid TENG (G-L TENG), liquid–liquid TENG (L-L TENG), etc. At present, the triboelectric sensors that are widely used are mainly focused on S-S TENG sensors [12,13,14,15,16,17] and L-S TENG sensors [18,19,20,21,22,23]. The S-S TENG originated from contact electrification (CE) between two solid materials. Because of the triboelectric effect, charges are generated at the contact interface due to the difference in electronegativity between the two materials. When the two materials are separated, owing to the electrostatic induction effect, opposite charges are induced on the corresponding electrodes to form a current. The self-powered sensors based on solid–solid CE can realize pressure sensing [24,25], tactile sensing [26,27], object recognition [28,29], motion prediction [30,31,32,33], etc., injecting impetus into the development and progress of human–computer interaction (HMI). Compared with S-S TENG, the L-S TENG has a more stable and durable output. The collection and utilization of environmental energy were realized by using the contact between water droplets and solid surfaces to generate electric energy [34]. Sensors based on liquid–solid CE have unique advantages in chemical sensing [35,36], as well as environmental sensing [37,38,39,40], and offer an excellent strategy for electronic skin [41].

TENG is based on CE from different materials, which provide a wide selection of possibilities. TENG with rich material selection can collect complex and diverse electrical signals, which are not only voluminous but also contain various characteristics related to amplitude, frequency, waveform, etc. In practical applications, multi-channel TENG sensors are used for data acquisition to improve robustness. This increases the number and complexity of data. It is worth noting that TENG sensors are usually vulnerable to some external influences, such as background noises, intensity changes, unwanted CE, etc., which increases the difficulty of data analysis. Therefore, it is difficult to manually process data containing weak and diverse features. AI technology can be used as a means of data analysis [42]. Machine learning (ML) [43] is an AI technology applied to data processing, which has powerful data handling capabilities and provides accurate prediction models. As an emerging technology for extracting nuances and processing multi-channel signals, ML provides a unique advantage for efficient analysis and processing of triboelectric signals [44,45]. Currently, widely used ML algorithm models in TENGs include support vector machine (SVM) [46], random forest (RF) [47], K-nearest neighbor (KNN) [48], convolutional neural network (CNN) [49], recurrent neural network (RNN) [50], artificial neural network (ANN) [51], etc. ML algorithms help TENG sensing systems reduce the influence of environmental factors and improve the sensing performance. For example, a low-pass filter was used to remove high-frequency noise [52]. It helped filter out the effects of external vibration or electromagnetic interference, making the TENG signal more stable. The key frequency components in the TENG signal were extracted by using the Fourier transform [53], which selected the most representative spectral features to reduce the sensitivity to external noise. Environmental parameters (such as temperature and humidity) were integrated into the ML models to correct the TENG output [54]. The key differences in ML algorithms applied to S-S TENGs and L-S TENGs are mainly reflected in the differences in data characteristics and sensing tasks. The data generated by the S-S TENG sensor include force, voltage, current, etc. It is suitable for detecting mechanical motion. The data characteristics of L-S TENG involve liquid motion, charge transfer, and changes in liquid properties. It is suitable for various liquid monitoring tasks. No algorithm can be specific to a certain type of TENG. When selecting the ML algorithms, it is essential to optimize and adjust according to the specific perceptual tasks and data features. For example, the characteristics of L-S TENG sensors are sometimes very subtle and hidden in the output signal. It is necessary to select a ML method that can extract features in depth. In addition, special attention should be paid to the influence of humidity on the signals. The suitable ML algorithm could be selected to classify and process the data for different TENG intelligent sensing systems, such as tactile sensing [55], gesture sensing [56], electronic tongue [57], etc. It is of great significance for the development of HMI [58,59,60,61], human health monitoring [62,63,64], intelligent sports [65,66], and many other fields.

The self-powered sensing system consists of energy collection/storage, sensing, monitoring, interaction, and other components. The combination of TENG sensors and ML has injected vitality into the intelligent self-powered sensing system. Selecting appropriate ML methods for data processing and target recognition according to different signal characteristics collected by TENG is a challenge. It is undeniable that the application of ML is very dataset specific. When selecting ML algorithms for S-S TENG and L-S TENG, it is essential to consider the specific characteristics of the interaction medium, the types of signals generated, and the intended sensing tasks. Adaptation and optimization of ML models based on these considerations can lead to more effective and accurate sensor systems. It is unrealistic to uniquely correspond an algorithm to an application domain. In this paper, we summarize the latest advances in ML algorithms assisting solid–solid TENG and liquid–solid TENG sensors, finding that the sample size and data characteristics play an important role. Therefore, as the complexity of the dataset increases, the intelligent self-powered sensing systems assisted by different ML models are presented. They may not be the only choice but can provide a reference for researchers. Firstly, the working mechanisms of S-S TENGs and L-S TENGs are reviewed. Furthermore, based on different data characteristics and target task requirements, the latest research progress and applications of various ML algorithm-assisted, self-powered sensors are discussed, as shown in Figure 1. Finally, ideas are provided for the future development of the combination of ML and TENG sensors.

## 2. Working Mechanism of TENG

### 2.1. Working Mechanism of S-S TENG

The working mechanism of S-S TENG is based on the coupling of the triboelectric effect and electrostatic induction [77]. The core components of S-S TENG are two kinds of solid materials with different electronegativities. When the contact state of the two materials changes, equal and opposite charges will be generated on the contact surfaces. The charge symbol on both sides depends on the electronegativity of the two solid materials. It tends to produce positive triboelectric charges on the side of the material that loses electrons and generates negative triboelectric charges on the other side. After the two solid materials are separated from each other, the triboelectric charges generate an induced potential difference at the electrodes on both sides. It drives electron transfer to the external circuit, resulting in continuous alternating current (AC) output. At present, according to the device structure and operation mode of S-S TENGs, it can be divided into four working modes, including contact–separation mode [78], sliding mode [79], single-electrode mode [80], and free-standing triboelectric-layer mode [81]. The S-S TENGs based on the triboelectric effect can not only convert external mechanical energy into electrical energy but also act as self-powered sensors. In recent years, the S-S TENG sensors have been widely used in security identification [82,83,84,85], tactile sensing [86,87,88], motion analysis [89,90,91], etc. For example, Mu et al. developed an electro-mechanosensory finger that achieved remote control and tactile sensing [92]. Based on a TENG array composed of the liquid metal-polymer conductive layer, the contact voltage signals of different objects were collected to realize the identification of 18 materials under different contact forces. Zhang et al. demonstrated an AI toilet that integrated triboelectric pressure sensing and image sensing [93]. It realized human health monitoring by using the pressure distribution of users’ different sitting postures.

### 2.2. Working Mechanism of L-S TENG

The long-term mechanical durability will be affected by the continuous CE of the two solid materials [94]. In addition, the performances of S-S TENG are greatly influenced by environmental factors, such as humidity and atmospheric pressure [95]. These issues can be addressed by changing from solid–solid to liquid–solid CE, which involves the triboelectric layers participating in CE composed of solids and liquids. Compared to S-S TENG, the output of L-S TENG is stable and durable, with advantages in efficiency, cost, and lifespan. For the solid–liquid contact process, a “two-step” model of electrical double-layer (EDL) formation was proposed by Wang et al. [96]. For example, when liquid contacts with a fluorinated ethylene propylene (FEP) surface, the overlap of electron clouds between molecules in the liquid and atoms on the FEP surface leads to electron transfer. This step only occurs when the liquid first contacts the FEP. Ion transfer happens in the second step, where cations migrate to the negatively charged FEP surface due to the electrostatic interaction, forming the EDL. L-S TENGs have also been classified into four working modes: contact–separation [97], lateral sliding [98], free-standing [99], and single electrode [100]. Recently, L-S TENGs can potentially be used as self-powered sensors in chemical detection [101,102,103,104], environmental monitoring [105,106,107], etc. For example, Liu et al. reported a chemical sensing probe that can measure the sucrose concentration based on the electron transfers of solid–liquid CE in different dissolution states of sucrose [108]. Wang et al. proposed a solid–liquid–solid structure TENG, which used the EDL to transmit energy and signals [109]. The device successfully converted the output voltage into an international Morse code and decoded the state of the finger in real time on the mobile phone screen.

## 3. ML for TENGs

Compared with other sensors, the signals generated by TENG sensors may have unique spectrum and waveform characteristics. Therefore, the time domain and frequency domain characteristics of the TENG signal need to be considered when selecting the algorithms. In addition, TENG sensors are usually used for energy harvesting and sensing tasks. In the case of limited energy, it is more practical to choose lightweight, low-power machine learning algorithms. When selecting the algorithm for the wearable TENG sensor, the requirements of real-time performance and small model size should also be considered. Whether it is S-S TENGs or L-S TENGs, the wide range of material selectivity and simple structure of the devices enable them to continuously collect a large amount of sensing data in a short period [110]. However, it is difficult to process these data directly without AI. To improve the analysis and processing of massive sensing data as well as sensing accuracy, ML has been widely employed as a tool to assist TENG sensors. Traditional ML methods include SVM, KNN, RF, etc., which can be used to deal with small-scale datasets [111]. Deep learning (DL) is a sub-field of ML as an extension of traditional ML. Including CNN, RNN, ANN, etc., DL is more suitable for large-scale datasets [112]. Traditional ML methods require manual design and extraction of features to convert raw data into a format suitable for the model. It is difficult to make use of the advantages of big data by relying on the prior knowledge and parameter adjustment experience of designers. DL can automatically learn features through the network, reducing the need for manual intervention [113]. Moreover, DL models use multi-layer neural networks, and each layer is capable of learning feature representations at different abstract levels of the data. It makes DL highly effective in handling massive multidimensional data. Although DL has such high performance, it does not mean that it universally applies to all situations. For smaller datasets, traditional ML algorithms are often superior to deep networks. It has stronger interpretability, making it easier to adjust parameters or change model designs. For TENG sensors, most of the output data that need to be analyzed and processed are electrical signals. Therefore, the considerations of electrical signal characteristics and target tasks are very important.

### 3.1. ML Algorithms for Small Datasets

#### 3.1.1. SVM for TENGs

When analyzing and processing small datasets, that is, only a relatively small number of samples or data points can be collected, SVM can be employed to assist TENGs in performing sensing tasks and avoiding overfitting. In the 1960s, Vapnik et al. began exploring the basic concepts of SVM and proposed the maximum margin principle for linear classifiers. In 1995, his team published a paper formally introducing the SVM algorithm [114]. The fundamental idea behind SVM learning is to find several data points on the margin and use them to define a plane (decision plane) that maximizes the distance to support vectors. Before the popularity of DL, SVM was the most popular classical model due to its robust mathematical foundation and the ability to achieve a global optimum. SVM is suitable for structured data types and was initially designed for binary classification [115]. It can cleverly handle nonlinear classification problems in high-dimensional space by introducing a kernel function. SVM proves effective in various tasks for TENG sensors, particularly in the classification of electrical signal data with distinct separation boundaries.

##### SVM for S-S TENGs

In scenarios with a limited number of samples featuring distinguishable characteristics, SVM holds a significant advantage. Therefore, it can assist the TENG sensors in achieving simple object and character recognition [116,117,118]. Recently, Zhao et al. designed an untethered triboelectric patch for intelligent sensing and energy harvesting [67]. The patch is primarily composed of polytetrafluoroethylene (PTFE) triboelectric layers and was attached to human skin, utilizing the human body as a conductor for TENG. As shown in Figure 2a(i), the operational cycle of the patch involved two phases: contact and separation. The triboelectric effect generated negative charges on the surface of PTFE during the contact process, resulting in a current when the human surface potential exceeded the ground (Figure 2a(ii)). The current direction reversed during the separation process (Figure 2a(iii)). The research showed that the sensor patch on the finger collected a total of 660 samples, which were randomly divided into a training group (58.4%) and a test group (41.6%). Combined with the SVM algorithm, it can effectively distinguish 11 objects, achieving an accuracy of 93.09–94.91%. In 2020, Ji et al. successfully engineered a TENG with remarkable sensitivity to recognize handwriting signals [119]. The detailed design of the smart TENG is shown in Figure 2b(i). The triboelectric layers in the TENG comprised a meticulously woven copper mesh and a NaCl-molded polydimethylsiloxane (PDMS) film. Throughout the handwriting process, the dynamic contact and separation between PDMS and the copper mesh facilitated electron transfer, generating discernible electrical signals (Figure 2b(ii)). Handwritten signals of 26 letters were collected, with a total of 520 samples. The t-distributed stochastic neighbor embedding was applied to reduce the dimension of the letter fingerprint to seven dimensions and used as features for further ML analysis. Combined with the SVM model, the data acquisition and processing of the writing signals for 26 letters resulted in a handwriting recognition accuracy of 93.5%.

For two-class or multi-class high-dimensional tasks, such as user recognition [122,123] and working condition recognition [124], etc., SVM exhibits absolute advantages. SVM can be integrated with principal component analysis (PCA) [125]. After feature extraction and dimension reduction by PCA, user authentication and recognition can be realized by the classifier. Wu et al. devised a keystroke dynamics security system leveraging distinct typing patterns among users [120]. The structure of a single triboelectric key is shown in Figure 2c(i), and the proof-of-concept triboelectric numeric keypad is displayed in Figure 2c(ii). A single triboelectric key consisted of a shielding electrode and a TENG employing the contact–separation mechanism. The sensitivity of the triboelectric keystroke device resulted in varied signal outputs for different finger sizes and typing forces. A user classification software platform was constructed based on the PCA algorithm and a customized SVM model based on the LibSVM toolbox [126] (Figure 2c(iii)). For an exemplary number sequence consisting of six digits, “8-0-7-3-4-5”, 17 features could be obtained. A total of 150 sets of data for each of the 5 users were randomly selected for training and testing. It achieved a remarkable 98.7% accuracy in verifying and precisely identifying user identity through the analysis of unique typing behavior. In addition to the keystroke dynamics device for user identification, in 2021, Zhou et al. drew a bioinspired, TENG-driven, self-powered sensor [68], enabling accurate user recognition through ML. Integrating a motion module and a TENG module, the structural design of the sensing system is shown in Figure 2d(i). The TENG module comprised a silver nanowires (AgNWs)-based bottom electrode, a silver nanowires/barium titanate nanoparticles/polydimethylsiloxane (AgNWs/BaTiO3NPs/PDMS) composite triboelectric layer, and a carbon-based top electrode. The contact and separation of TENG were analogous to the contraction and relaxation of muscles, resulting in corresponding current signals. A total of 15 features were collected after 4 triggers. Similarly, PCA was employed for feature extraction and dimensionality reduction, and a two-class SVM classifier constructed a database for decision-making (Figure 2d(ii)). With the assistance of ML, the user successfully obtained system authorization after four triggers, while the intruder who imitated the user to trigger four times was denied access. Based on different vibration state data of the vibration TENG, machine faults can be detected. As shown in Figure 2e(i), Li et al. collected mechanical vibration energy by using a multi-layer vibration triboelectric nanogenerator (V-TENG) and established a multi-node sensor network to identify the different working states of the machine [121]. Three self-powered vibration sensor node (SVSN) modules based on V-TENG were placed on the working machine to obtain vibration information. The radar plot of the data received of four working conditions/faults is shown in Figure 2e(ii). The multi-classification SVM classifier was used to train and test the 500 sets of data samples under 4 working conditions (Figure 2e(iii)). Finally, the recognition accuracy of different vibration states of the machine reached 83.36%, achieving fault detection.

##### SVM for L-S TENGs

Compared with other sensing mechanisms, the characteristics of the output signals of L-STENG are diverse. Therefore, it is challenging to quantitatively characterize its physical parameters directly. With the development of AI, an increasing number of next-generation sensors integrate data acquisition with signal processing to improve the efficiency and accuracy of sensing. As the most classical traditional ML method, SVM has been employed not only to assist S-S TENGs but also in studies on L-S TENGs. With the assistance of SVM, L-S TENGs can effectively monitor and identify liquid leaks. Zhang et al. designed a liquid leakage detection and identification device based on a single-electrode liquid–solid (SELS) TENG [69], as shown in Figure 3i. Using p-type silicon wafer as a solid triboelectric layer, the system realized high-sensitivity detection of liquid leakage. When liquid droplets continuously fell and came into contact with the liquid remaining on the SiO_2_ surface, the triboelectric effect and electrostatic induction phenomenon produced a continuous and periodic current flowing from the ground to the copper electrode (Figure 3ii). The SELS TENG self-powered sensors could be installed at locations such as pipe joints, valves, and flanges, where liquid leaks are most likely to occur (Figure 3iii). The frequency of the short-circuit output current can be used to qualitatively characterize the liquid leakage rate. Additionally, using PCA dimensionality reduction and the SVM-based ML model, an intelligent detection and recognition system could be established to distinguish different types of liquids (Figure 3iv). After using the k-fold cross-validation (K-CV) method to optimize the parameters, the system achieved a good performance, exceeding 90% classification accuracy for each of the two liquids on 120 data samples.

In summary, SVM can be an effective tool in TENG sensors, especially excelling in handling high-dimensional spaces and/or small datasets. Compared to some other algorithms, SVM performs well even with a relatively small number of training samples, demonstrating robustness against overfitting. However, SVM may become computationally intensive when dealing with large-scale datasets, requiring significant memory and time resources. This might limit its practical use in certain sensing applications. At the same time, careful parameter tuning (for example, selecting the kernel function and setting regularization parameters, etc.) is necessary to achieve optimal performance. These limiting factors prompt further exploration of more suitable ML algorithms in diverse application scenarios to enhance the performance of TENG sensing systems.

#### 3.1.2. KNN for TENGs

In scenarios with small-scale datasets where the distribution of signal data is irregular in space, it may be challenging for SVM to give a decision plane. In such cases, KNN [127], as an instance-based learning method, might offer a more straightforward solution. KNN classifies or regresses based on the majority class of the k-nearest neighbors around a new data point, making it less influenced by data distribution. The KNN model is simple and easy to implement and is often employed for classification tasks [128]. After selecting the appropriate k-value range, it can achieve high-precision recognition while mitigating the impact of individual outlier signals. Therefore, for some simple problems and small-scale datasets where the neighbor relationship among data points needs to be considered, KNN can be a preferred choice.

In addition to considering the information within individual signals, handwritten signals also require an understanding of their relationship with neighboring signals, making KNN a suitable choice for simple handwritten recognition. Guo et al. reported an intelligent HMI based on TENG [129]. They utilized TENGs with a horizontal–vertical symmetric array of electrodes to record the triboelectric signal sequences of different handwriting trajectories. The triboelectric signal of handwritten characters based on triboelectric nanogenerators is shown in Figure 4a(i). When the slider came into contact with the PDMS surface, PDMS became negatively charged. As the slider moved across the electrodes, negative charges on the PDMS surface were gradually screened, leading to a current flow from the electrodes to the ground. Subsequently, as the slider moved away from the electrode, positive charges were induced on the PDMS corresponding to the electrode, generating a current from the ground to the electrode. Combined with PCA dimensionality reduction and the KNN algorithm model (Figure 4a(ii)), the system successfully identified handwritten English letters, Chinese characters, and Arabic numerals, while eliminating interference from individual abnormal noise signals (Figure 4a(iii)). Moreover, the applications of the KNN models to assist TENGs in speech recognition have also been explored. In 2020, Liu et al. introduced a self-powered artificial auditory pathway driven by TENG [70], effectively distinguishing speech commands in a noisy environment. The artificial auditory pathway consisted of TENG and field effect synaptic transistor (FEST), with its structure shown in Figure 4b(i). The contact electrification process induced by the speaker’s sound wave applied a negative gate voltage on the synaptic transistor due to charge transfer (Figure 4b(ii)). The acoustic signal recognition process by the artificial auditory pathway is shown in Figure 4b(iii). Each word records 40 groups of data, of which 20 groups were used as the training database and the other 20 groups were used as the test database. Utilizing KNN as the classifier, after eight rounds of training, the speech recognition accuracy for the seven-word instructions successfully reached 95%. In the selection of the KNN algorithms, it is essential to choose an appropriate k-value range. In 2023, Liu et al. presented a method to enhance the triboelectric electrification of marine polysaccharides and developed a flexible, flame-retardant, and environmentally friendly TENG sensor [130]. As shown in Figure 4c(i), the sensor used alginate fibers and vermiculite (VMT) nanosheets as triboelectric materials to construct a vertical contact–separation mode triboelectric nanogenerator (CS-TENG). To capture the clustering features of the target motion, PCA was performed using the three extracted features. After dimensionality reduction with PCA and classification using KNN with k = 2 (Figure 4c(ii)), the predictive accuracy of 9 different motion types reached 96.2%.

As an algorithm that stores data in the training phase and calculates in real time in the prediction phase, KNN excels at completing real-time tasks. Moreover, KNN can directly add new data points for classification, while SVM needs to retrain the model. Therefore, KNN is well-suited for applications that require adaptation to new data in real-time and rapid predictions. Qiu et al. reported a bionic, mechanical, dual-modal sensor capable of real-time pressure sensing and display in a pattern [71]. The mechanical sensor adopted a single-electrode-mode TENG, and its triboelectricity–mechanical luminescence scheme is shown in Figure 4d(i). It can visualize the writing trajectory in real time and reflect the strength of the writing (Figure 4d(ii)). Based on this, researchers developed an electronic skin array for HMI. The KNN algorithm was utilized to effectively identify and classify a set of training data (numbers 0–9) with an accuracy of 96.92% (Figure 4d(iii)). Traditional ML methods, such as SVM and KNN, demonstrate excellent performance in handling limited data and demand fewer hardware resources. However, in large-scale sensing systems, it is usually necessary to process more complex data types. Traditional ML methods require additional feature engineering, whose training speed, memory usage, time, and other factors constrain the applications in more complex sensing tasks. Therefore, the integration of DL methods with TENGs can assist in accomplishing more intricate sensing tasks and adapting to a broader range of application scenarios.

### 3.2. DL Algorithms for Large-Scale Datasets

#### 3.2.1. ANN for TENGs

Traditional ML methods usually require feature engineering, which makes it difficult to take advantage of big data in triboelectric sensing [131]. In contrast, DL eliminates this step and it can also handle large volumes of multidimensional and multi-type data, adapting to various tasks [132]. Therefore, DL has become the preferred choice to assist TENGs in achieving target detection, image recognition, multifunctional sensing [133], etc. ANN was first proposed in the 1950s and formed the foundation of DL [134]. Inspired by biological neural networks, ANN is a mathematical model simulating information transmission among neurons in the brain. Its basic principle involves constructing multiple layers of neurons for information transmission and processing through weighted connections and activation functions [135]. ANN comprises input layers, hidden layers, and output layers, with each layer containing multiple neurons. The advantage of ANN lies in its ability to learn complex nonlinear relationships, making it suitable for various ML tasks, including classification, regression, clustering, etc.

##### ANN for S-S TENGs

When the features of the electrical signal data are not readily interpretable or designable, the sensing performance can be enhanced by utilizing ANN to learn effective features from the data. In 2022, Zhang et al. reported a wearable TENG gait analysis and waist motion capture device for improving the performance of lower limbs and waist rehabilitation [53], as shown in Figure 5a. The device included two triboelectric sensors for walking state detection. The ANN model combined with fast Fourier transform (FFT) was used to extract features from the triboelectric signals, eliminating time-domain information while retaining frequency characteristics to identify patients for rehabilitation plan selection. Each sensor performed 150 tests on each of the 5 participants to obtain a dataset. At the same time, the 400-point length signals were visualized using a specific window. Therefore, each sample has 400 × 2 = 800 features. Notably, the network structure with different fully connected layers (FCL) was adopted, and dropout and batch normalization layers were inserted to enhance robustness and accelerate the training speed. Ultimately, the 5-FCL-based ANN model architecture exhibited the best predictive performance, achieving an overall recognition accuracy of 98.4%. Due to its versatility, ANN is also extremely important for multimodal sensing systems. Pang et al. developed a multifunctional tactile sensor [136], as shown in Figure 5b. This tactile sensor has been proven capable of recognizing speech and real-time monitoring of physiological signals and human movement. Combining an ANN model consisting of one input layer, three hidden layers, and one output layer, the recognition of nine material types with different textures reached 94.44%. Specifically, all neurons between each layer are fully connected, and the input time domain signal of each material has 400 neurons.

Developing multifunctional and diverse artificial neural systems to integrate multimodal plasticity, memory, and supervised learning functions is a crucial task in neuromorphic computing simulation. Because artificial synapses coincide with the original intention of ANN design, the use of artificial synaptic models in artificial neural networks is also a very common application. In 2021, Yu et al. designed a biomimetic, mechanical, photonic artificial synapse combining a phototransistor and a TENG, as shown in Figure 5c(i) [72]. While simulating the synaptic function (Figure 5c(iii)), high-precision image recognition could be achieved by constructing an ANN (Figure 5c(ii)) assisted by mechanical plasticity. With 6 × 105 synapses and an increase in the number of training samples to 60,000, the highest recognition accuracy reached 92%. An example of the mapping image obtained from ANN is shown in Figure 5c(iv). Moreover, inspired by the retina, Guo et al. proposed an optoelectronic synaptic device based on polyimide (PI)/graphene heterostructures [137]. The basic structure and function are shown in Figure 5d(i). The bidirectional dynamic control of synaptic weights was achieved through the photoelectric coupling modulation of TENG and ultraviolet light (Figure 5d(ii)). Furthermore, an analog ANN was further constructed for handwritten digit recognition. In the absence of a hidden layer, an accuracy of 84% was achieved in less than 2100 training cycles. The digit “8” at initial and final states is shown in Figure 5d(iii).

##### ANN for L-S TENGs

S-S TENGs may encounter limitations on stretchability and flexibility when applied to soft electronic systems. The introduction of hydrogel into TENGs to form liquid–solid CE can effectively solve the above problems [138]. The previously mentioned ANN can assist TENGs in gait analysis. Hydrogels are also used in flexible sensors for gait analysis due to their stretchable, self-healing, and biodegradable properties. Wang et al. developed a self-powered strain sensor based on graphene oxide-polyacrylamide (GO-PAM) hydrogel [139], as shown in Figure 6a. The sensor served both as a TENG for harvesting mechanical energy and as part of a wearable insole monitoring system. Various algorithms were employed to recognize normal and pathological gaits, with ANN achieving the highest recognition accuracy of 99.5% and 98.2%, respectively. ANN models can also contribute to constructing self-powered sensing systems for L-S TENGs used in marine environmental monitoring. In 2022, Wang et al. reported a wave-driven L-S TENG [54], which was constructed with an ethylene chlorotrifluoroethylene (ECTFE) film and an ionic hydrogel electrode for SO_2_ gas detection (Figure 6b(i,ii)). As shown in Figure 6b(iii), the working principle of the system was based on a sliding L-S TENG. Since ANN has advantages in sensor error correction, the model was selected and three input samples of temperature, humidity, and voltage response were set as input layers. After repeated learning and correction, the effective monitoring of the marine environment was realized, with the final error being less than 3%.

In conclusion, ANN serves as the foundational framework for DL, capable of approximating various functions. However, its performance is often suboptimal when dealing with complex data structures, such as images and sequential data of TENGs. Therefore, specialized neural network architectures, such as CNN for spatial data and RNNs for sequential data, have been designed based on ANN to address these specific challenges.

#### 3.2.2. CNN for TENGs

The signals may be coupled with a variety of information, making it challenging for a simple neural network to distinguish and extract specific features, especially when dealing with complex information types, such as images. In such cases, building neural networks to handle more intricate data structures becomes necessary. CNN is a specialized variant of ANN that is designed for processing images and spatial data and proves invaluable in these scenarios. As one of the representative algorithms in DL, the basic concept of CNN dates back to the 1960s and 1970s. However, the real development and widespread application occurred in the early 21st century [140]. The success of CNN is attributed to advancements in computational power, the availability of large-scale datasets, such as ImageNet, and improvements in DL algorithms. It has become a crucial foundation in computer vision and DL, achieving remarkable accomplishments in various fields [141]. The basic structure of a CNN includes layers, such as the input layer, convolutional layers, pooling layers, fully connected layers, and the output layer. The core idea of a CNN involves the hierarchical extraction and combination of features through multiple layers of convolution and pooling operations. CNN’s capability to gradually extract multi-level feature representations allows it to efficiently construct abstract representations, starting from simple to complex features. The convolutional operations in CNN effectively reduce the number of model parameters, improving the training speed and generalization capability [142,143]. Therefore, by combining the CNN algorithm with TENGs, efficient high-precision sensing can be achieved for tasks such as image recognition [144,145,146], gesture recognition [147,148], material identification [149,150,151], etc.

##### CNN for S-S TENGs

The original purpose of CNN is to process image data, making it indispensable in scenarios where triboelectric signals contain image features. Yun et al. proposed a triboelectricity touchpad (TTP) featuring a TENG array built on a flexible substrate [73], as shown in Figure 7a(i). The data acquisition board collected the triboelectric data from each row and column, determining positions generating electric signals (Figure 7a(ii)). This facilitated the inference of handwritten digits on the TTP. A pre-trained neural network was employed for handwritten digit recognition on the TTP, with the CNN comprising input, hidden, and output layers, along with convolution and linearization layers (Figure 7a(iii)). Following training with 60,000 data points, the classification accuracy of handwritten digits ranged from 93.6% to 91.8% at bending angles of 0°, 119°, and 169°. In 2022, Yang et al. developed a multifunctional wearable tactile sensor [152], as shown in Figure 7b(i), combining TENG and the piezoelectric nanogenerator (PENG) in a six-layer structure. The sensor attached to the hand sensed deformation and output electrical signals during various hand gestures. Based on a convolutional neural network with six convolutional layers, three pooling layers, a connection layer, and a dropout, a deep learning framework was built (Figure 7b(ii)). Convolution and pooling were employed to extract high-level features from images, reducing feature maps to one-dimensional vectors fed into the fully connected layer. After 1000 iterations of training and testing, an accuracy of 94.16% was achieved. To extract more abstract features when processing images, the visual geometry group (VGG), one of the representative networks of CNN, is applied to triboelectric sensing [153]. In 2022, Wei et al. developed an intelligent tactile sensing system [154], which realized the accurate perception of test materials under different contact conditions and external environments. As shown in Figure 7c, the system consisted of a triboelectric triple tactile sensor (TTS) array and a CNN-based DL model. The total dataset (containing 6365 sets) used for training came from different touch and environmental conditions of nine materials. By extracting features from three single tactile sensors and normalizing them, a material recognition system using image extraction feature points was built. The high recognition rate of 96.62% was achieved in the material recognition of nine kinds of materials.

CNN excels at extracting subtle features hidden in signals, providing an effective approach for recognizing and processing signals that are challenging to distinguish directly. For the multi-channel TENG systems, CNN proves to be an efficient aid. To achieve comprehensive and continuous monitoring of infants, Guo et al. proposed a triboelectric hydrogel sensor [155], as shown in Figure 7d(i). A single sensor consisted of 3 layers, agar hydrogel, and gelatin sponge (Figure 7d(ii)), with 11 sensors attached to the baby’s breasts, hands, knees, feet, neck, back, wrists, and buttocks, forming an 11-channel data acquisition system. CNN was used to rapidly process signals generated by multiple channels and identify the different motion states of the baby (Figure 7d(iii)). CNN has been demonstrated to analyze various types of sensing data. In 2022, Fang et al. developed a breathable and moisture-proof TENG sensor [156], as shown in Figure 7e(i). The textile sensor was composed of two types of twisted triboelectric yarns: PVDF and epoxy resin (Figure 7e(ii)). It collected respiratory signals from common respiratory cases and combined a 1D-CNN algorithm to achieve accurate identification of different breathing patterns for respiratory diagnosis (Figure 7e(iii)). In 2023, Zu et al. proposed a sensor composed of a fully 3D-printed, multi-angle TENG (MA-TENG) to realize effective monitoring of head impact [157], as shown in Figure 7f(i). The structure and working principle are shown in Figure 7f(ii). The array consisted of 32 MA-TENG units, and each MA-TENG unit operated in single-electrode mode. Then, 180 data points were recorded for each channel signal data input. To accurately assess head impacts, a deep convolutional neural network (DCNN) was employed (Figure 7f(iii)), achieving a high accuracy of 98% in damage level evaluation.

##### CNN for L-S TENGs

Small but important characteristics are hidden in the output signals of different solution types or concentrations after CE. The convolution operation of CNN can realize automatic learning and local extraction of these features. Based on prior research efforts, the L-S TENGs can utilize CNN models for tasks such as classification and recognition [158]. In pursuit of low-cost, real-time sediment detection, Yu et al. reported a droplet-driven triboelectric nanogenerator (PLDD-TENG) [159]. As shown in Figure 8a(i), the mechanism included liquid–PTFE contact electrification and particle–electrode electrostatic induction. When the droplets fell on the pre-charged PTFE, the EDL was formed to generate a positive current, and then the droplets rebounded to generate a negative current. Utilizing the current signals from PLDD-TENG, a 1D CNN model was employed to identify sediment types and mass fractions (Figure 8a(ii)). This method achieved high-precision identification of two types of tasks and developed a dual-target recognition system that can monitor suspended sediment parameters in real time. Similarly, capitalizing on the sensitivity of L-S TENG output signals to particle types and concentrations, Huang et al. devised a structurally straightforward L-S TENG for identifying and detecting microplastics in liquids [160], as shown in Figure 8b. It utilized FEP film and liquid as triboelectric materials. Upon contact of the FEP film with water, a negative charge layer was formed on the FEP surface. The reciprocating motion of L-S TENG caused charge transfer to generate electrical signals. It was observed that the output voltage signals of the L-S TENG were sensitive to the particle types and mass fractions of microplastics in the liquid, exhibiting a good linear relationship with the content of microplastics. Each dataset was randomly divided into two groups: 161 groups for training and 60 groups for testing. Before training, the collected data must be normalized. Combining a CNN model consisting of one input layer, three convolutional layers, three pooling layers, and one fully connected layer, the type and content of each microplastic were predicted. The TENG sensor achieved a higher average recognition accuracy rate (86.7%), especially with a 100% recognition accuracy for polystyrene (PS).

Based on liquid–solid CE, TENGs present unique advantages in chemical sensing. Recently, Wei et al. proposed a triboelectric taste sensing system based on the dynamic morphological changes of droplets and the principle of liquid–solid CE [57]. This system efficiently sensed liquids by detecting the difference in charge transfer between the liquid and the solid surface. The sensing system is shown in Figure 8c. The differences in electron affinity and physicochemical properties of different liquids led to distinct triboelectric signals caused by the charge transfer triggered on the electrode. The total dataset used for robotic taste sensing was from 5 different liquids, containing 52,586 samples (training set, 47,330). Through the combination of CNN, a “liquid fingerprint database” was established to achieve high accuracy (91.3%) in robot taste applications. Further integration of visual information with sensory signal information was performed to extract complete visual information about the liquids. This improved the perception capabilities of the taste system, increasing the accuracy to 96%. To enhance the stability of triboelectric electric sensors, quantifying sensor performance through the number of wave peaks rather than the output amplitude has been recognized as an effective method. Ge et al. developed a flexible microfluidic triboelectric electric sensor (FMTS) that utilized the flow of liquid in microfluidic channels to generate output signals [162]. The structure and mechanism of the FMTS are shown in Figure 8d(i,ii). It consisted of a flexible PDMS substrate with microfluidic channels and two chambers, cross-shaped ITO electrodes, and a PDMS film serving as the triboelectric layer. The squeezing of the finger when the finger was bent led to the difference in the output signals, and the multidimensional feature extraction of the output waveforms was carried out (Figure 8d(iii)). The sensors were attached to 5 fingers, respectively, and each sensor signal recorded 200 data points. A total of 500 samples were collected, and each gesture was 100. Utilizing a CNN model, accurate recognition of five hand gestures was achieved with a precision of 99.2% (Figure 8d(iv)). Furthermore, based on the waveform characteristics of the sensor under dynamic pressure, a coding system was established, achieving a recognition accuracy of 98.8% for eight types of information.

In summary, CNN is effective in extracting local features and reducing computational complexity through parameter sharing. It enhances TENG’s performance in image recognition, target detection, etc. However, CNN has limitations in effectively handling sequential data due to its lack of memory capacity for sequence information. Therefore, to process the time series data of TENG and achieve dynamic monitoring and recognition, the introduction of RNN is necessary.

#### 3.2.3. RNN for TENGs

The sensing data sometimes contain temporal information, which can be challenging for CNN to effectively capture and process. Therefore, another ANN variant, RNN, was proposed to analyze and process the triboelectric signals containing temporal information [163]. RNN is suitable for describing outputs in continuous states, exhibiting memory capabilities [164]. In RNN, neurons not only receive information from other neurons but also themselves, forming a network structure with loops. By utilizing internal states (hidden states) to retain previous information, RNN can capture contextual information, making it more effective than traditional neural networks in handling sequential data. However, traditional RNN suffer from issues such as vanishing gradients and exploding gradients. Subsequently, improved structures, such as gated recurrent units (GRU) and long short-term memory networks (LSTM) [165], were developed to further enhance the performance of RNN in time series tasks.

When dealing with time series data, RNN is good at capturing the sequence relationship. In 2022, Lu et al. proposed a lip-reading decoding system based on a flexible TENG sensor [75], as shown in Figure 9a(i). The sensor employed the contact–separation mode of the dual-electrode TENG structure, to monitor the change in the oral state (Figure 9a(ii)). There was no charge change in the closed state of the mouth. When the mouth opened, the nylon membrane came close to the PVC, generating a current from the PVC side electrode to the nylon side electrode. An extended RNN model based on the prototype learning method was presented, with GRU chosen as the basic unit of the recurrent neural network (Figure 9a(iii)). The test accuracy of 20 kinds of words reached 94.5% in training 20 classes with 100 samples each, providing convenient life assistance for people with speech impairments.

In RNN, each output is determined by both the current input and the preceding information, which is a clever approach but comes with a problem. When the sequence is very long, the issue of vanishing gradients arises. This means the weights of neurons in the early layers hardly change, leading to ineffective training. LSTM addresses it by replacing traditional neurons in the RNN architecture with input, output, and forget gates [168], learning long-term-dependent information easily [124,168,169]. Ran et al. developed a real-time blood pressure monitoring system using a TENG with a double-sandwich structure [76]. As shown in Figure 9b(i), the system operated a silicone rubber film and cardboard to form an outer interlayer, and a Cu film and cardboard to form an inner interlayer, which flexibly responds to various scenarios in a single-electrode mode. By capturing pulse signals at the radial artery, a multi-network structure consisting of an LSTM layer, an attention layer, a convolution layer, and a fully connected layer was developed for blood pressure estimation (Figure 9b(ii)). The mean absolute error and standard deviation of error were 3.79 ± 5.27 and 3.86 ± 5.18 mm Hg, respectively. The LSTM model has also been applied in capturing non-driving behaviors. Zhang et al. proposed a real-time non-driving behavior recognition system [170]. As shown in Figure 9c(i), the triboelectric sensor data and image data of five types of non-driving behaviors of eight drivers were collected and analyzed using a clever single-electrode structure. To achieve accurate recognition, two multi-class classifiers based on artificial feature extraction and three multi-class classifiers based on neural network feature extraction were trained and tested. Among them, LSTM (Figure 9c(ii)) had the highest test accuracy (93.5%). Building upon LSTM, Bidirectional LSTM (BiLSTM) [171] has been introduced. In BiLSTM, the input sequence is processed simultaneously in both forward and backward directions, generating two sets of hidden states: one composed of forward information and the other composed of backward information. This bidirectional structure enables the network to capture long-term dependency relationships and contextual information within the sequence more effectively. Li et al. developed a TENG-based gait sensing unit for monitoring human activities and identifying users [166], as shown in Figure 9d(i). They designed a Residual Dense-BiLSTM network to extract deep features from multi-channel temporal gait data (Figure 9d(ii)). The multi-layer stacked structure helped address the problem of multi-class time series gait recognition and improved the recognition performance. After 500 iterations, the sensing system achieved accuracy rates of 97.9% for human behavior recognition and 99.4% for user identity recognition. To achieve efficient extraction of temporal and spatial features simultaneously, Mao et al. introduced a hybrid CNN–BiLSTM–Attention model into a wearable TENG sensor to detect and analyze the limb movement of patients with Parkinson’s disease [167]. As shown in Figure 9e(i), the electrical signals were obtained by the contact and separation processes of nylon and PET. Local features of the signal were extracted by CNN, and then the BiLSTM model was employed to learn the long-term dependence between features. Following the bidirectional LSTM layer, an attention mechanism was added to automatically capture the most relevant features in the input sequence (Figure 9e(ii)). In the present dataset, 532 training samples (80%) and 228 test samples (20%) were used for each action. Finally, the recognition accuracy of eight postures reached as high as 97.3%.

In conclusion, the strength of RNN lies in its capacity to capture dependencies in TENGs’ sequence data, leading to extensive applications in fields such as natural language processing and speech recognition. LSTM, as an improved version of RNN, successfully addresses issues such as vanishing and exploding gradients present in traditional RNN, exhibiting enhanced memory capacity and an improved capability to capture long-term dependencies. Therefore, RNN and LSTM are well-suited for handling sequence data of varying lengths and complexities in time series tasks, holding significant relevance for the dynamic monitoring functionality of TENG sensors.

### 3.3. Comparison of Key Parameters

The evaluation of key parameters of ML algorithms in some TENG sensors has been summarized, such as numbers of training data (the number of data samples in the training set), training epochs, accuracy, etc., to provide suggestions for different self-powered sensors. To show them more intuitively, a comparison table (Table 1) was constructed. For TENG sensors with small sample collection, traditional ML algorithms, such as SVM and KNN, can achieve high-precision sensing. When the larger sample data could be obtained by TENG sensors, the neural networks were selected more often in the past research.

## 4. Conclusions and Prospects

As the datasets collected become increasingly complex, different approaches to ML and DL algorithms have been proposed to efficiently process output signals from self-powered TENG sensing systems. In this review, suitable scenarios for various algorithms are proposed in Figure 10. This summary aims to assist future research on TENG-based sensors, guiding making faster and more effective choices when selecting ML algorithms.

In summary, when dealing with TENG sensors that can only gather a limited amount of sample data, traditional ML methods, such as SVM and KNN models, exhibit superior performance. Specifically, SVM proves effective in improving the sensing accuracy when dealing with highly distinctive sample data. Additionally, if there is a complex nonlinear relationship between the signals and the sensing tasks, SVM can flexibly handle nonlinear mappings through kernel functions, enhancing its ability to model complex relationships. However, challenges arise when the triboelectric signals exhibit irregular distribution in space, making it difficult for SVM to identify a suitable decision surface. In such cases, KNN was usually selected by researchers. Furthermore, KNN is a lazy learning algorithm, which only stores data during the training phase and conducts real-time calculations during the prediction phase. If real-time processing is a crucial requirement for sensing tasks, KNN might be a feasible choice.

When a substantial amount of data is accumulated, traditional ML methods might fail. In this case, DL algorithms, such as ANN, CNN, and RNN, become essential. Serving as the foundation for DL, ANN proves valuable when the features of triboelectric data are challenging to interpret or design. In addition, since ANN is a general neural network, it plays a pivotal role in assisting TENG sensing systems in integrating multimodal data. However, when the triboelectric signals contain complex data structures, such as images or time series data, CNN and RNN are more concerning to researchers. CNN can automatically learn and extract local features through convolution operations. If the performance of the TENG sensors is influenced by specific local environments, it can adeptly capture the local features. It is essential to note that if triboelectric sensing data involve image information, CNN might be the preferred choice. Furthermore, when the data are temporal, RNN is proficient in capturing sequential relationships in the time series data. To overcome the gradient disappearance, GRU and LSTM have been introduced to improve the RNN. In conclusion, when choosing ML algorithms to enhance the accuracy of sensors based on S-S TENGs or L-S TENGs, it is essential to consider the data characteristics and system requirements to ensure the algorithm model matches the task requirements.

To meet the requirements of the development of the IoT for large-scale sensing systems, more and more algorithms will be integrated with TENGs. Currently, self-powered intelligent sensing systems based on TENGs are primarily confined to laboratory research, and external or internal disturbances in non-ideal environments will affect the sensing performance of the systems. Faced with a multi-field coupling environment, using a single S-S TENG or L-S TENG, it is challenging to collect all the signal data. Moreover, the existing single ML methods struggle to decouple and process complex data. Therefore, the prospects of integrating ML algorithms with TENGs to construct new generations of sensing systems may include:Improve the data acquisition capability of TENGs. Develop G-S, G-L, L-L, or composite TENG structures [174,175]. Hybrid nanogenerators can collect high-quality and more comprehensive signals under complex environmental conditions. The improvement of the data acquisition capability ensures that ML algorithms achieve better learning effects in training and testing. Additionally, since ML is highly dependent on data, it is essential to develop TENGs with stable output to guarantee data quality. This can be achieved by selecting triboelectric materials or structures with better durability [176].Optimize the algorithms. When the existing algorithms cannot meet the deployment requirements of large-scale triboelectric sensors, new ML algorithms can be developed based on the specific data characteristics of TENGs. On the other hand, the integration of multiple DL algorithms, such as using multi-modal information [177] or multi-task learning [178] methods, can improve the data processing ability of the system. In addition, the algorithm models should learn human environmental perception, emotional preferences, and the ability to avoid disadvantages. Reinforcement learning [179] is an effective strategy to adapt to dynamic environmental conditions by cultivating the interaction between agent and environment to learn the best decision.Multi-domain applications. Knowledge in different fields can provide more optimization schemes for TENG sensing systems. Intelligent sensing systems gain more knowledge reserves and key technologies in different human activities, which help the machine to more comprehensively imitate the perception, thinking, decision-making, and collaboration capabilities of the human brain [180].Optimize energy harvesting. The energy harvesting functionality of TENGs can also benefit from algorithmic assistance [181]. Utilizing ML to optimize the energy management of triboelectric sensors enables more efficient energy harvesting and utilization. This optimization enhances the stability and sustainability of the sensor, reducing energy waste.

The advancement of integrated circuits and embedded technology allows computer systems to be integrated into a variety of devices. It facilitates the design, testing, and deployment of hardware and software collaboration, and improves the efficiency of intelligent sensing systems. ML-assisted TENG sensors will continue to focus on technologies such as HMI and brain–computer interfaces to transform machines from passive output to active creation. In the future, intelligent sensing systems will broaden the scope of applications, including smart homes, smart cities, smart healthcare, smart manufacturing, etc.

## Figures and Tables

**Figure 1 nanomaterials-14-00165-f001:**
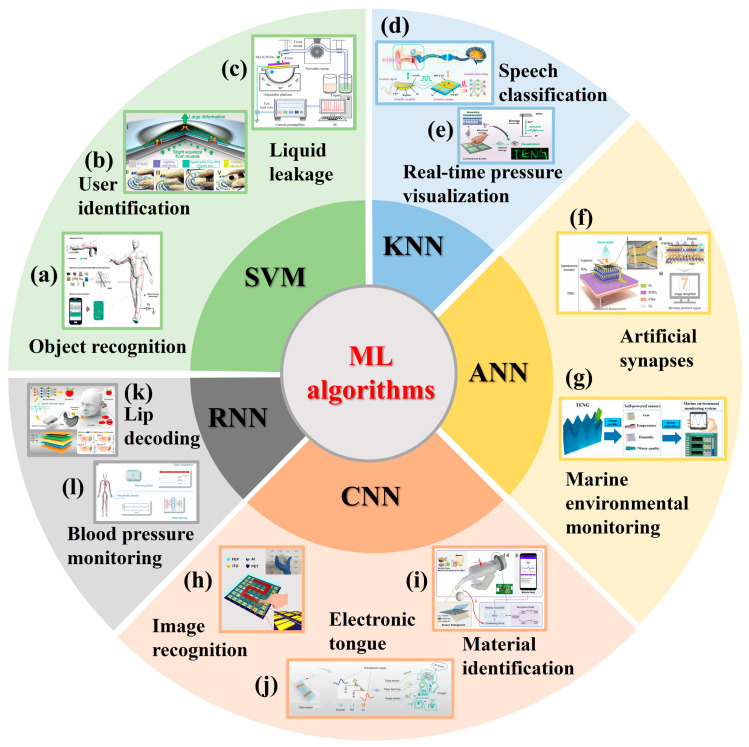
ML algorithm-assisted TENG sensors. (**a**) TENG sensor for object recognition [67]; (**b**) TENG sensor for user identification [68]; (**c**) TENG sensor for liquid leakage monitoring [69]; (**d**) TENG sensor for speech classification [70]; (**e**) TENG sensor for real-time pressure visualization [71]; (**f**) TENG sensor for artificial synapses [72]; (**g**) TENG sensor for marine environmental monitoring [54]; (**h**) TENG sensor for image recognition [73]; (**i**) TENG sensor for material identification [74]; (**j**) TENG sensor for taste sensing [57]; (**k**) TENG sensor for lip decoding [75]; (**l**) TENG sensor for blood pressure monitoring [76].

**Figure 2 nanomaterials-14-00165-f002:**
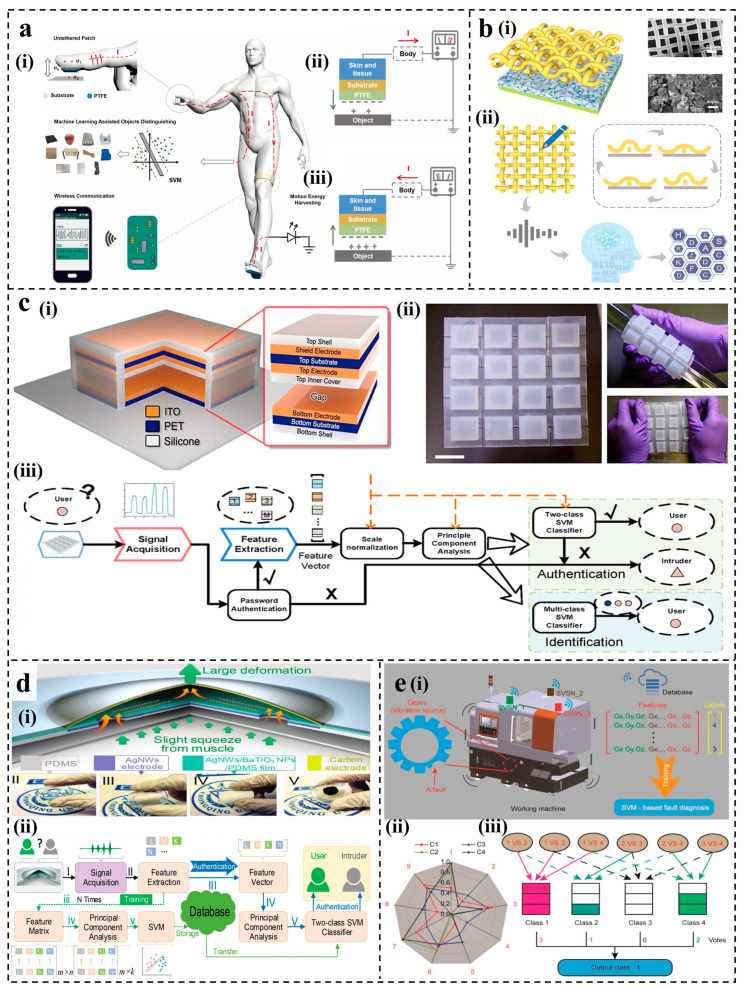
Applications of SVM for S-S TENGs. (**a**) Untethered triboelectric patch for wearable smart sensing [67]. (**b**) Design of a copper-mesh-based triboelectric nanogenerator for identification of 26 letters [119]. (**c**) Keystroke dynamics-based security system for identifying users [120]. scale bar = 2 cm. (**d**) Bionic principle, structure, and working mechanism of the BTUSE sensor [68]. (**e**) Machine fault detection based on a multi-node, self-powered sensor network [121].

**Figure 3 nanomaterials-14-00165-f003:**
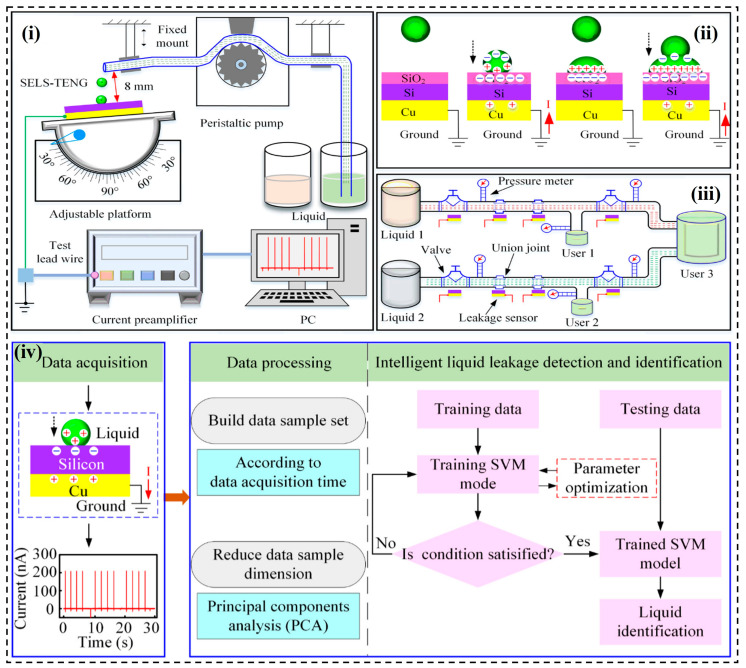
Applications of SVM for L-S TENGs. The test platform, working mechanism, and potential practical application of the SELS TENG-based self-powered liquid leakage detection sensor [69].

**Figure 4 nanomaterials-14-00165-f004:**
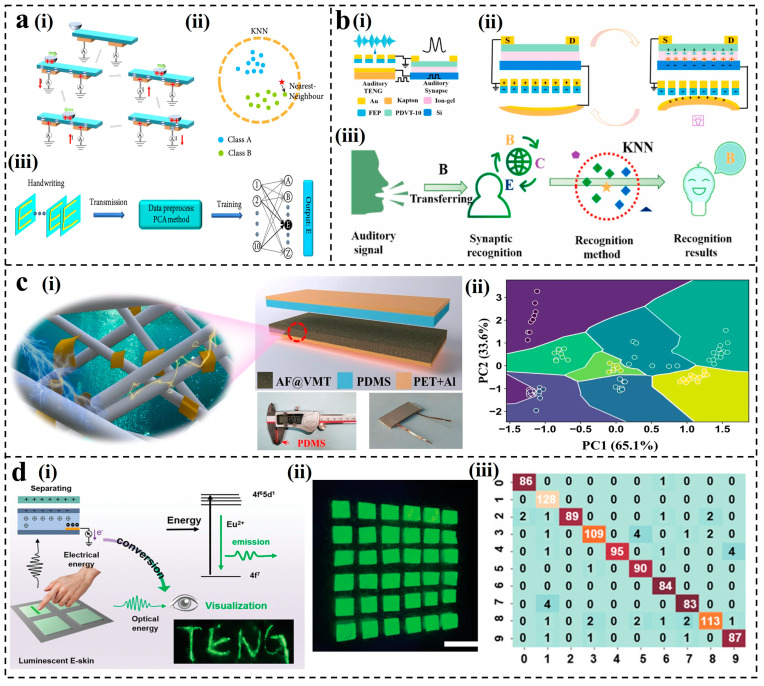
Applications of KNN for TENGs: (**a**) Schematic of the human–machine interface based on an intelligent handwriting recognition system [129]. (**b**) Self-powered artificial auditory pathway for sound detection [70]. (**c**) Phyllosilicate-polysaccharide triboelectric for human motion prediction [130]. (**d**) The working principle and application of the bioinspired bimodal mechanosensors [71]. scale bar = 10 mm.

**Figure 5 nanomaterials-14-00165-f005:**
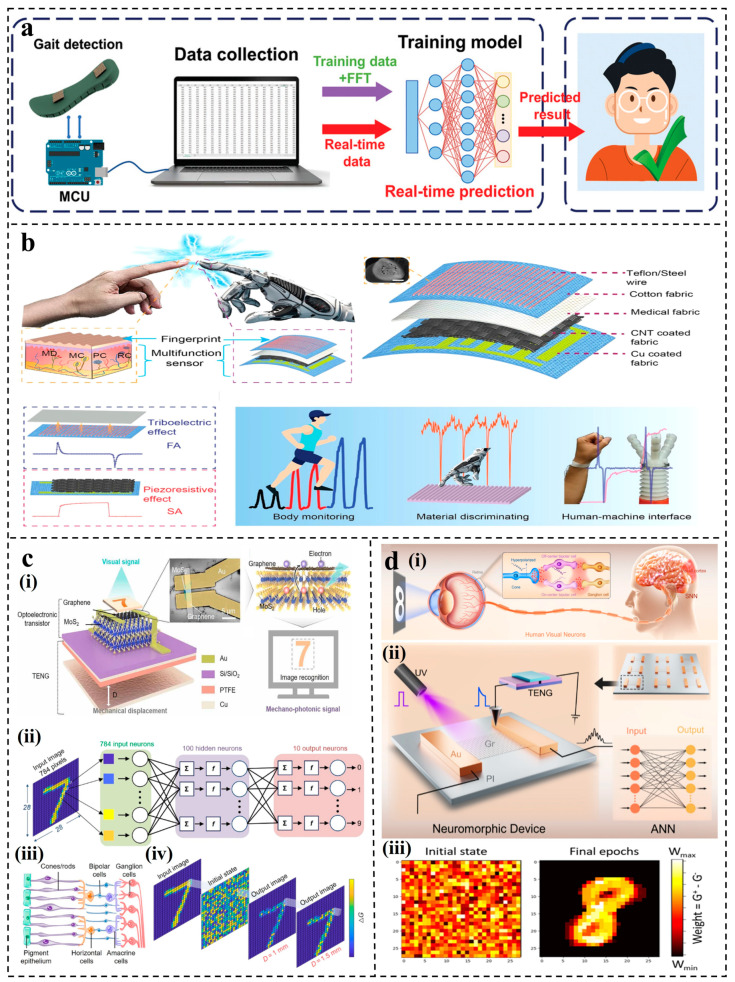
Applications of ANN for S-S TENGs: (**a**) Wearable triboelectric sensors for IoT-based smart healthcare [53]. (**b**) Textile tactile sensors for multifunctional sensing [136]. scale bar = 100 μm. (**c**) Bioinspired mechano-photonic artificial synapse [72]. (**d**) Retina-inspired neuromorphic optoelectronic device [137].

**Figure 6 nanomaterials-14-00165-f006:**
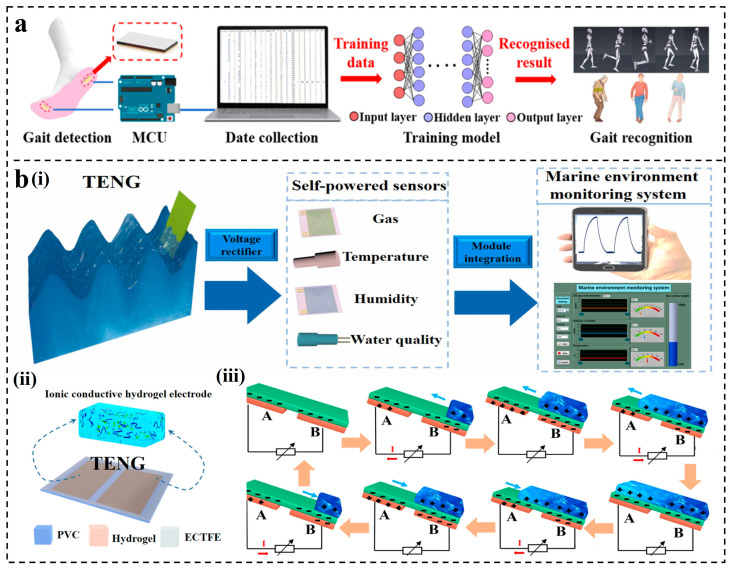
Applications of ANN for L-S TENGs: (**a**) A triboelectric smart sensor based on graphene oxide and polyacrylamide hydrogel [139]. (**b**) Ethylene chlorotrifluoroethylene/hydrogel-based L-S TENG [54].

**Figure 7 nanomaterials-14-00165-f007:**
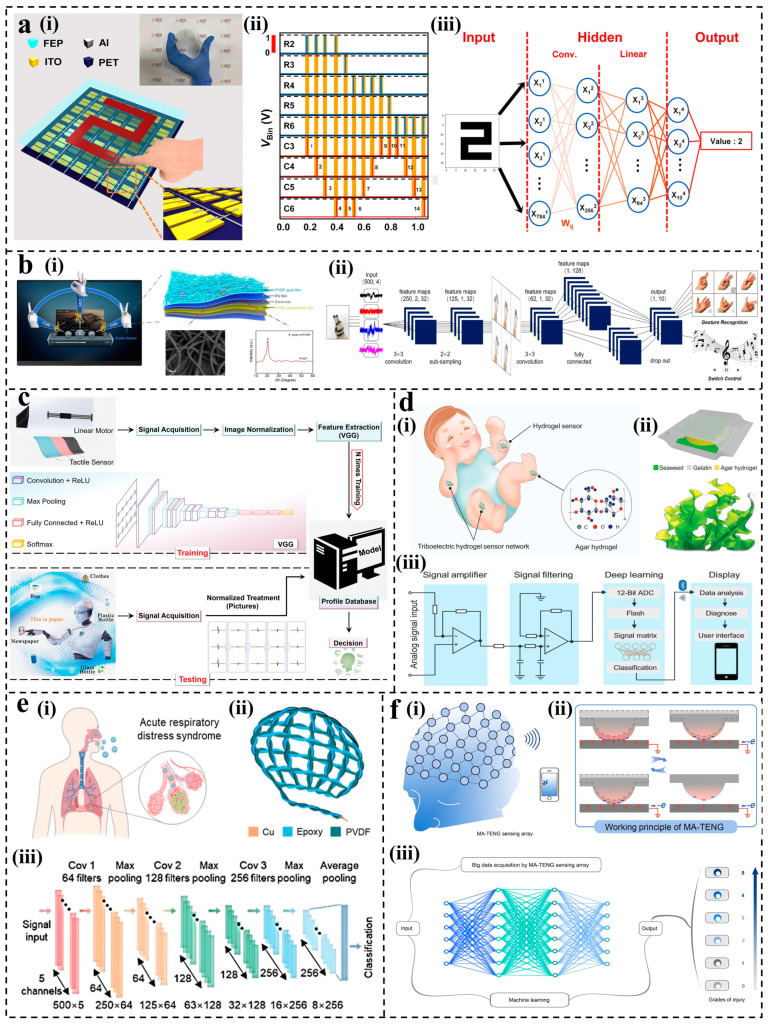
Applications of CNN for S-S TENGs: (**a**) The schematic illustration of the proposed touchpad and the pre-trained neural network [73]. (**b**) Design of an intelligent system for gesture recognition of the tactile sensor [152]. (**c**) The enhanced material-identification system using a neural network model (VGG) [154]. (**d**) Design of a soft, edible triboelectric hydrogel sensor for infant care [155]. (**e**) Design of the respiratory monitoring system [156]. (**f**) Multi-angle, self-powered sensor array for real-time monitoring of head impact [157].

**Figure 8 nanomaterials-14-00165-f008:**
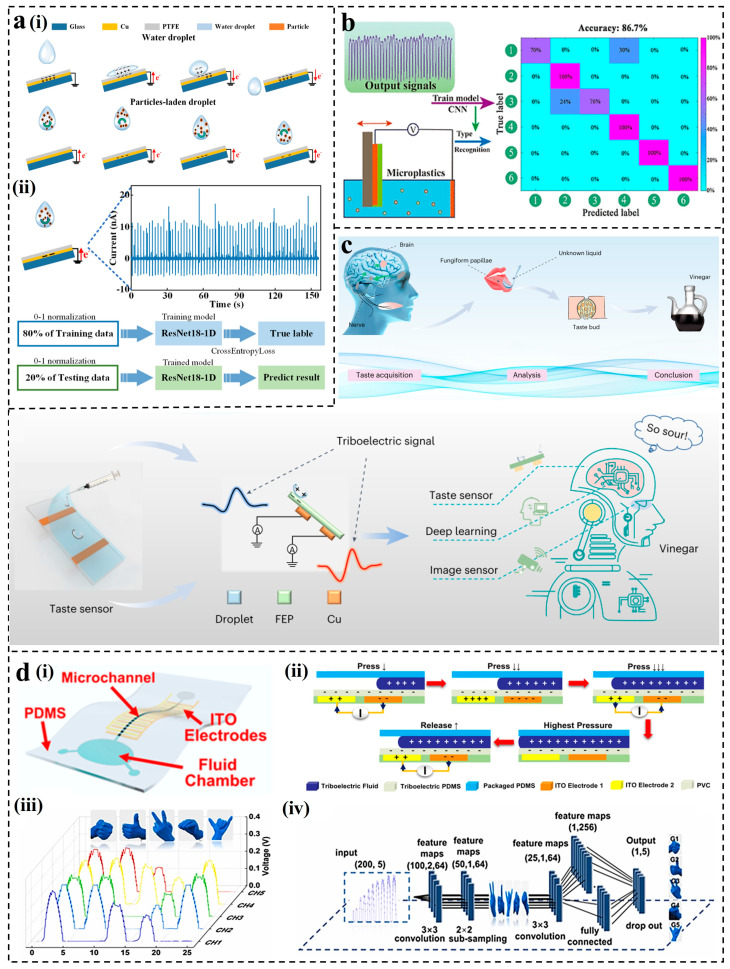
Applications of CNN for L-S TENGs: (**a**) Real-time monitoring of sediment particle parameters [161]. (**b**) Detection of microplastics based on L-S TENG [160]. (**c**) Droplet-based triboelectric taste-sensing system mimicking the human taste receptor [57]. (**d**) Demonstration of the FMTS as an angle sensor for gesture recognition [162].

**Figure 9 nanomaterials-14-00165-f009:**
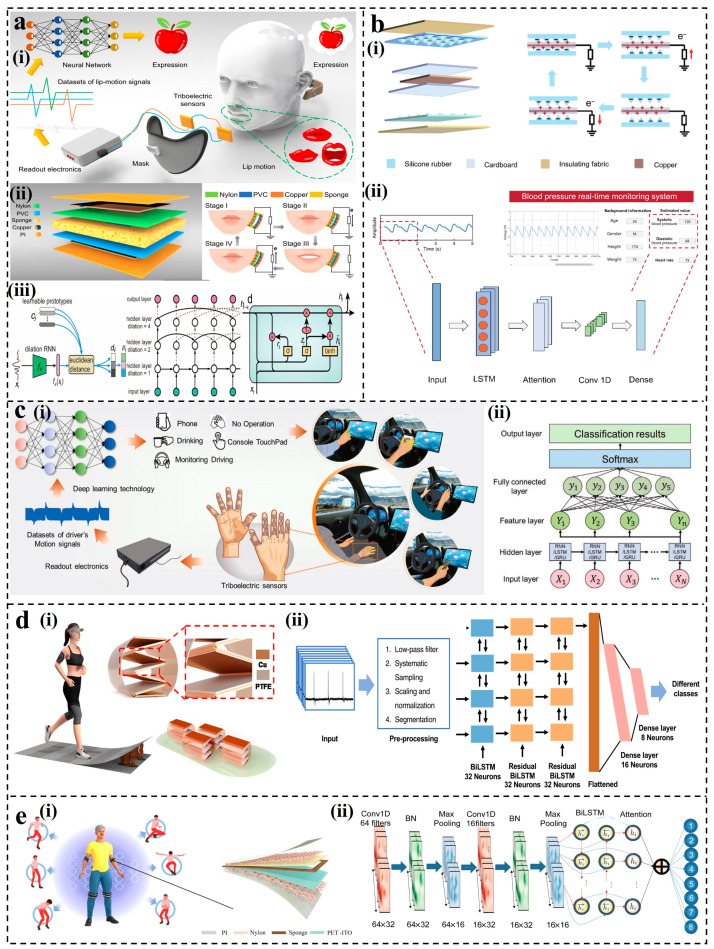
Applications of RNN for TENGs: (**a**) The lip-language decoding system supported by triboelectric sensors [75]. (**b**) Blood pressure monitoring system with a double-sandwich structure [76]. (**c**) The concept, structure, and mechanism of the real-time non-driving behavior recognition system aided by triboelectric sensors. (**d**) System overview of the TENG-based gait sensor system for human activity recognition and user identification [166]. (**e**) Overview of the intelligent IoMT monitoring system and detailed sensor structure design [167].

**Figure 10 nanomaterials-14-00165-f010:**
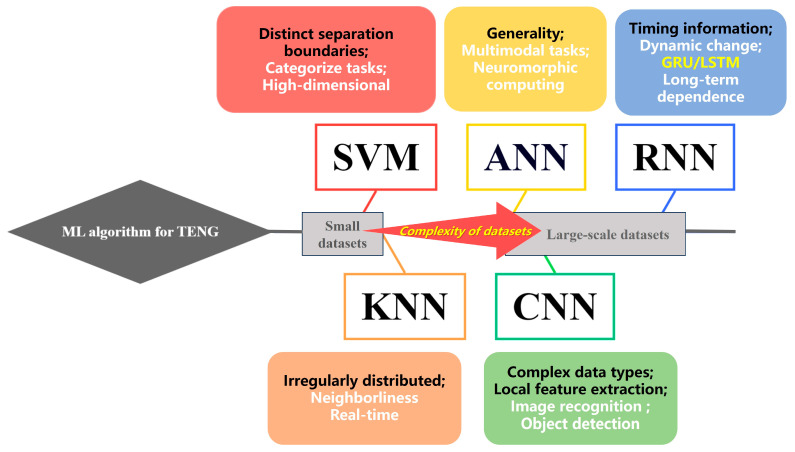
ML algorithms for TENGs with the complexity of datasets.

**Table 1 nanomaterials-14-00165-t001:** Comparison of key parameters of ML algorithms in TENG sensors.

ML Algorithms	Numbers of Training Data	Training Epochs	Accuracy
SVM	120	\	90% [69]
	260	\	93.5% [119]
	300	\	98.9% [123]
	385	\	94.91% [67]
	1500	\	83.6% [121]
KNN	140	8	95% [70]
	56/315	\	98.2%/100% [66]
ANN	1000	10	98.4% [53]
	6480	20	94.44% [136]
CNN	800	50	96% [82]
	3920	50	99.07%/99.32% [74]
	6365	100	96.62% [154]
	47,330	3	91.3% [57]
	60,000	1000	96.83% [73]
	311,950/128,000	100	98.50%/98.3% [172]
RNN	2000	500	94.5% [75]
	4256	\	97.3% [167]
	5000	100	81.06% [173]

## Data Availability

No new data were created or analyzed in this study. Data sharing is not applicable to this article.

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
