# Peer review of "Synergizing Machine Learning Algorithm with Triboelectric Nanogenerators for Advanced Self-Powered Sensing Systems"

_nanomaterials, 2024, doi:10.3390/nano14020165_

Round 1
Reviewer 1 Report
Comments and Suggestions for Authors
Dear authors,
I would like to express my sincere appreciation for the excellent work done in the review. The integration of triboelectric nanogenerators (TENGs) with machine learning (ML) and deep learning (DL) algorithms for large-scale intelligent sensing systems in the IoT era was comprehensively and insightfully addressed. The in-depth analysis of the state of the art, with exceptionally recent and relevant references, demonstrated meticulous commitment and significantly enhanced the credibility and relevance of the review. His work is undoubtedly a valuable and up-to-date source for the community of researchers and developers in this field. I highly recommend publishing the review in its current form, which not only reflects a solid knowledge of the topics discussed, but also demonstrates dedication to excellence in research.
Reviewer 2 Report
Comments and Suggestions for Authors
Abstract:
- Lines 11-12: authors could be more specific about examples of what is actually used for batteries right now and level of environmental risk. Human staffing costs associated with replacement are also important.
- Line 15: unclear what datasets are being referred to here. Also not clear how this concept follows logically – are the authors suggesting nodes that run ML algorithms on site and only send back post-processed data?
Manuscript:
- Lines 29-31: this is true, but it is not clear that this manuscript addresses this challenge in any way
- Line 56-58: unclear what is being stated here and why the data would be so complex. Is this because it is envisioned to have multiple of this kind of sensor? Or because response for a single sensor is nonlinear?
- Figure 1 – missing citations for all examples included in the figure.
- Lines 140-142: “Traditional ML methods usually require additional feature engineering, which transforms raw data into a feature vector. So, the accuracy largely depends on the quality of feature engineering.” This statement is not generally true. Authors should provide citations as well as correct language to be more nuanced.
General comments:
- The authors are attempting to map ML algorithms onto domain areas, however in reality the application of ML is very data-set specific (not domain specific). The analysis lacks a clear understanding of why different algorithms work in different cases – and a discussion on whether they were truly the “only” or “best” option (or whether the study authors simply selected that approach without testing others).
- The analysis lacks a clear explanation for how decision making related to algorithm selection would be different for data from these sensors than for any other type of sensor, and there are many other review papers published covering such analyses. It is not clear how the ML/AI component of this paper is intended to go beyond information already available in the literature.
- It is implied by this manuscript that “self-powered sensors” could be enabled, but it is not clear whether this is suggesting that the same device be used for sensing and for generating power. It would seem this would have inherent conflicts. Could the proposed hardware configuration be described in more detail? This was a major motivation highlighted in the abstract and the introduction but does not come through clearly in the manuscript itself.
Comments on the Quality of English Language
There is frequent mismatch of subject/verb throughout the manuscript.
Reviewer 3 Report
Comments and Suggestions for Authors
This manuscript offers a comprehensive review of the machine learning algorithms applied to triboelectric nanogenerators. It mainly introduces several key algorithms, including SVM, KNN, CNN, RNN, ANN, etc., which can help us understand the choice of the corresponding algorithm for specific condition. I would like to suggest minor revision with the following comments.
1. Can the authors give a bit more explanation about the reason of categorizing the TENG into S-S TENG and L-S TENG? Are there any key differences between the ML algorithms applied in S-S TENG and L-S TENG?
2. TENG sensors are usually vulnerable to some external influences, such as background noises, humidity changes, unwanted triboelectrifications, etc. In addition to the various algorithms used in TENG, the authors are suggested to provide some analysis regarding the recommendation of utilizing the algorithms to improve the performance of TENG sensors.
3. Some comparisons and evaluations of ML algorithms are suggested for the key parameters, like numbers of training data, training epochs, accuracies, etc., in order to give suggestions to different applications.
Comments on the Quality of English Language
The overall quality of english is good.
